# Clinical phenotypes of chronic cough categorised by cluster analysis

Jiyeon Kang[1], Woo Jung Seo[1], Jieun Kang[1], So Hee Park[1], Hyung Koo Kang[1], Hye Kyeong Park[1], Sung-Soon Lee[1], Ji-Yong Moon[2], Deog Kyeom Kim[3], Seung Hun Jang[4], Jin Woo Kim[5], Minseok Seo[6], Hyeon-Kyoung Koo[1]*

1 Division of Pulmonary and Critical Care Medicine, Department of Internal Medicine, Ilsan Paik Hospital, Inje University College of Medicine, Goyang, Republic of Korea, 2 Department of Internal Medicine, Hanyang University College of Medicine, Guri, Republic of Korea, 3 Division of Pulmonary and Critical Care Medicine, Department of Internal Medicine, Seoul Metropolitan Government-Seoul National University Boramae Medical Center, Seoul National University College of Medicine, Seoul, Republic of Korea, 4 Division of Pulmonary, Allergy, and Critical Care Medicine, Department of Medicine, Hallym University Sacred Heart Hospital, Hallym University College of Medicine, Anyang, Republic of Korea, 5 Division of Pulmonary and Critical Care Medicine, Department of Internal Medicine, Uijeongbu St. Mary's Hospital, College of Medicine, The Catholic University of Korea, Uijeongbu, Republic of Korea, 6 Department of Computer and Information Science, Korea University, Sejong, Republic of Korea

* gusrud9@yahoo.co.kr

## Abstract

### Background

Chronic cough is a heterogeneous disease with various aetiologies that are difficult to determine. Our study aimed to categorise the phenotypes of chronic cough.

### Methods

Adult patients with chronic cough were assessed based on the characteristics and severity of their cough using the COugh Assessment Test (COAT) and the Korean version of the Leicester Cough Questionnaire. A cluster analysis was performed using the K-prototype, and the variables to be included were determined using a correlation network.

### Results

In total, 255 participants were included in the analysis. Based on the correlation network, age, score for each item, and total COAT score were selected for the cluster analysis. Four clusters were identified and characterised as follows: 1) elderly with mild cough, 2) middle-aged with less severe cough, 3) relatively male-predominant youth with severe cough, and 4) female-predominant elderly with severe cough. All clusters had distinct demographic and symptomatic characteristics and underlying causes.

### Conclusions

Cluster analysis of age, score for each item, and total COAT score identified 4 distinct phenotypes of chronic cough with significant differences in the aetiologies. Subgrouping

**Data Availability Statement:** All relevant data are within the paper and its Supporting information files.

**Funding:** This work was supported by the National Research Foundation of Korea (NIRF) grant funded

by the Korean government (MIST: No. 2021R1G1A1095110). The funders had no role in study design, data collection and analysis, decision to publish, or preparation of the manuscript.

**Competing interests:** The authors have declared that no competing interests exist.

patients with chronic cough into homogenous phenotypes could provide a stratified medical approach for individualising diagnostic and therapeutic strategies.

## Introduction

Cough is one of the most common symptoms of pulmonary and extra-pulmonary diseases, leading patients to seek medical attention [1, 2]. The most common causes of chronic cough are known as upper airway cough syndrome (UACS), cough variant asthma (CVA), eosinophilic bronchitis (EB), and gastroesophageal reflux disease (GERD) [3–6]. However, the associated symptoms of chronic cough vary widely, and determining the aetiology of chronic cough is a difficult process as it can develop for various reasons [7–11]. Guidelines for the evaluation of chronic cough include a variety of specialized equipment that is impractical for many primary care centres due to limited resources [3–6]. Furthermore, the pathogenesis of this heterogeneity has not been fully established. Defining the subtypes of these heterogeneous phenotypes of chronic cough is important for better understanding the mechanisms of disease development and progression. Cluster analysis is an unsupervised machine-learning approach used to define specific phenotypes. This analysis has been employed for diverse diseases, including chronic obstructive pulmonary disease [12], congestive heart failure [13], and sepsis [14].

This study aimed to identify the specific subtypes of patients with chronic cough through cluster analysis. We hypothesised that the demographic and clinical signs of the patients would reflect the aetiology of chronic cough. To select the variables to be included in the cluster analysis, we initially searched for features associated with the causes of chronic cough using a correlation matrix. Thereafter, we used K-prototype clustering to categorise the phenotypes of patients with chronic cough and compare their demographics, symptoms, and causes between the clusters. Patients' symptoms and their severities were measured using the COugh Assessment Test (COAT) [15] and the Korean version of the Leicester Cough Questionnaire (K-LCQ) [16, 17].

## Materials and methods

### Study participants and ethics

Adult patients ($\geq$ 18 years old) with chronic cough lasting > 8 weeks were recruited from 16 respiratory centres in South Korea. All potential participants were enrolled between December 2016 and July 2017. The cause of chronic cough was assessed following the Korean cough guideline [18] by pulmonary specialists in each hospital, excluding those for patients with suspected abnormalities on plain chest radiography/computed tomography, or with chronic respiratory disease, such as overt asthma, COPD, bronchiectasis, tuberculosis-destroyed lung, and lung cancer. The enrolled participants completed both the COAT [15] and K-LCQ [16, 17] questionnaires at their first visit.

The COAT is a simplified version of the K-LCQ and is used to assess the severity of the cough. It consists of five factors: frequency of the cough (COAT 1), limitation of daily activities (COAT 2), sleep disturbance (COAT 3), fatigue (COAT 4), and hypersensitivity to irritants (COAT 5). All factors were scored on a single scale of 0–4 (total scores of 0–20), where a higher score indicates a more severe cough. We defined cough severity as mild, less severe, and severe if COAT scores corresponded to the first, second, and third to fourth quartiles, respectively. The K-LCQ is a validated cough-specific questionnaire for quality of life, containing 19

questions about physical, psychological, and social domains. A 7-point Likert scale was used for each question (range of 1–7), then mean values for each domain were calculated (scores of 1–7 for each domain), and the total scores were produced by summation of each domain with range of 3–21; higher score indicates a better quality of life. The physical domain is composed of questions regarding chest/stomach pain accompanied by bothersome phlegm, fatigue, hypersensitivity to irritants, sleep disturbances, frequency of coughing bouts, voice hoarseness, and loss of energy. In the psychological domain, questions regarding feeling discontent, worrying about serious illness, and concerns regarding what other people may think were included. The social domain includes questions regarding interference with jobs or daily tasks, life enjoyment, interruption of telephone conversations, and annoyance of partners, family, or friends. Consequently, the K-LCQ and COAT scores were highly associated with a negative correlation [6]. This study was conducted in accordance with the Declaration of Helsinki, and the institutional review board (IRB) of Inje University Ilsan Paik Hospital approved the study protocol (IRB No. ISPAIK 2017-12-025) and waived the need for informed consent as none of the patients were at risk.

## Statistical analysis

Patient characteristics were presented as the mean and standard deviation for continuous variables and as relative frequencies for categorical variables. A student's t-test was used to compare the two groups for continuous response variables such as age, COAT, and K-LCQ scores. For comparison between two categorical variables such as sex and comorbidities, the chi-squared test was used. All statistical analyses were performed using R Project software (version 3.6.0, https://www.r-project.org/). Correlation matrix was drawn for Pearson's r coefficient using the corrplot package (https://cran.r-project.org/web/packages/corrplot). Cluster analysis was performed using K-prototype clustering, including age, sex, and scores for COAT questions 1–5 and total COAT scores, which contributed to the specific cough phenotype in the correlation matrix and network. Continuous variables were assessed using a standard scale. The K-prototype was constructed using the clustMixType software package (https://cran.r-project.org/web/packages/clustMixType). The optimal number of clusters was selected based on average silhouette width. The silhouette value reflects how similar it is to its own cluster (cohesion) compared with other clusters (separation). The silhouette width was based on the pairwise difference of distances between and within the cluster to validate the performance of clustering. The optimal number of clusters was defined as when the index reached its maximum [19]. The value of the silhouette width was measured using the cluster R package. Finally, the consistency and reliability of clustering outcomes were evaluated based on the random forest classification model with 10-fold cross-validation.

## Results

### Baseline characteristics

A total of 255 patients with chronic cough (mean age, 47.7 ± 14.3 years) who completed both questionnaires with necessary diagnostic work-up were enrolled from 16 respiratory centres. Among them, 94 (36.9%) men and 161 (63.1%) women were included, with a male-to-female ratio of 1:1.71. The mean duration of cough was 18.3 ± 18.4 weeks. The mean score of the COAT was 11.3 ± 4.1 and that of the K-LCQ was 11.2 ± 3.1. The histogram of the COAT score is drawn in S1 Fig. As the causes of chronic cough, 116 patients (45.5%) were diagnosed with upper airway cough syndrome, 66 (25.9%) with asthma/cough-variant asthma (CVA), 41 (16.1%) with eosinophilic bronchitis, and 34 (13.3%) with gastroesophageal reflux disease

(GERD) (S2A Fig). In 16 (6.3%) patients, the cause could not be identified, and a total of 20 participants (7.8%) had multiple causes for their cough (S2B Fig).

## Correlation matrix

To understand the correlations between demographics, characteristics, and the severity and aetiologies of chronic cough, correlation matrices for COAT and K-LCQ scores were calculated (S1 Table). Since the COAT scores showed similar correlations with the causes of chronic cough compared to the K-LCQ scores, the COAT score was chosen as a variable for further analysis instead of the K-LCQ score because of its simple application in clinical practice. A correlation matrix (Fig 1) was created using the demographics and COAT scores, and only statistically significant correlations were drawn. Age was negatively correlated with the COAT scores for questions 1–5 and the total COAT scores, but female sex was positively associated with the COAT scores for question 4 and the presence of asthma/CVA. The presence of asthma/CVA was positively correlated with female sex and the COAT scores for questions 3 and 4. The presence of GERD was negatively associated with the COAT scores for questions 2–5 and the total scores. The scores for each question of the COAT were closely correlated with each other.

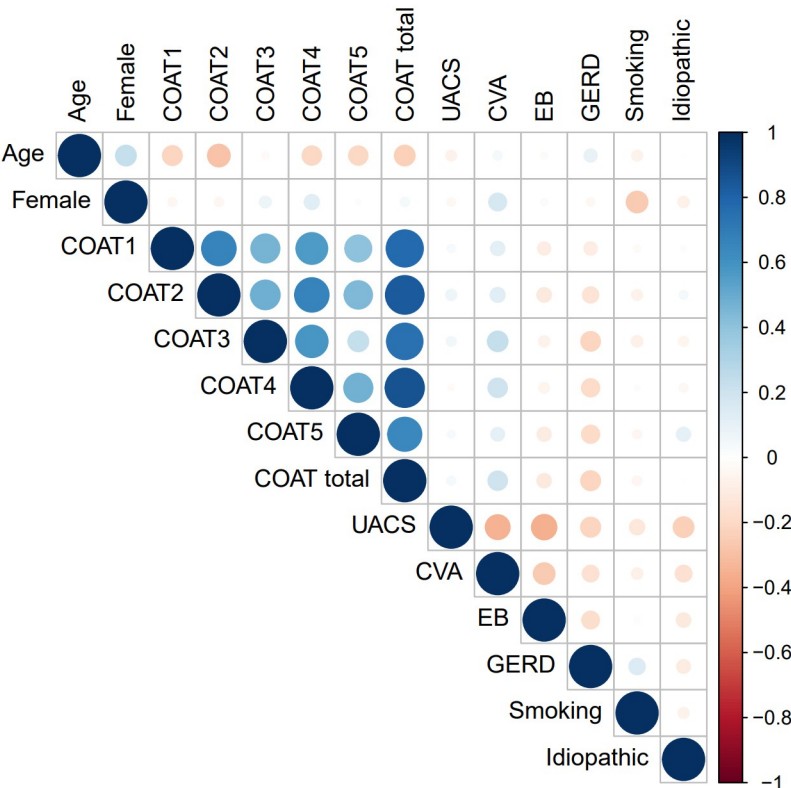

**Fig 1. Correlation matrix for COAT score and aetiologies of chronic cough.** Pearson's correlation between variables was performed. The intensity of the colour correlates with the strength of the association. Blue indicates positive correlation and red indicates negative correlation. COAT 1: frequency of cough, COAT 2: limitation of daily activities, COAT 3: sleep disturbance, COAT 4: fatigue, COAT 5: hypersensitivity to irritants.

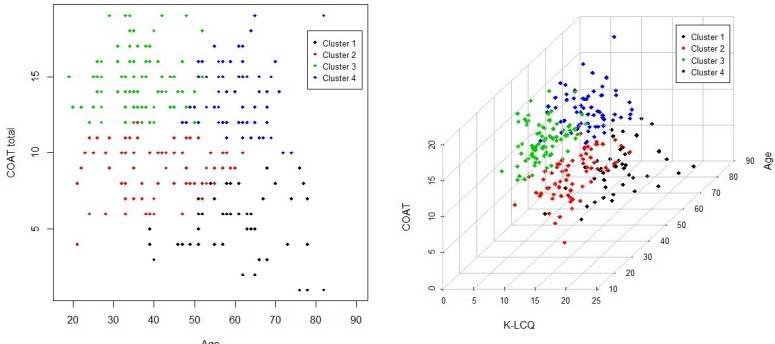

**Fig 2. Distribution of each of the 4 clusters by age and COAT score (A) and a 3-dimensional plot including the K-LCQ score (B).** (A) X and Y axis mean age and COAT total score (B) X, Y, Z axis mean K-LCQ, age, and COAT score, respectively. Cluster analysis was performed by the K-prototype clustering using the clustMixType package. Different colour indicates each cluster (black: cluster 1, red: cluster 2, green: cluster 3, and blue: cluster 4).

## Cluster analysis

Age, sex, COAT scores for questions 1–5, and total COAT scores were selected for cluster analysis based on the correlation matrix. Four clusters were determined using silhouette width (S3 Fig). The distribution of each cluster by age and total COAT score as well as a 3-dimensional plot including K-LCQ scores are shown in Fig 2A and 2B. Advanced age was negatively correlated with the total COAT score in the overall sample (coefficient = -0.07, P < 0.001); however, this significance was not found within each cluster (S4 Fig). The differences in characteristics between the 4 clusters are compared in Table 1 and S2 Table. The duration of the cough did not differ between these clusters.

Clusters were designated as follows: 1) elderly with mild cough, 2) middle-aged with less severe cough, 3) relatively male-predominant youth with severe cough, and 4) female-predominant elderly with severe cough (Fig 3). The radar charts comparing the patterns of each COAT question for each cluster are summarised in Fig 4. The proportions of each cause of chronic cough according to cluster are summarised in Fig 5. Random forest analysis with 10-fold cross-validation was technically performed to validate the consistency of the clustering results. The accuracies for each cluster 1–4 were 0.980 (SD: 0.031), 0.975 (0.037), 0.994 (0.018), and 0.985 (SD: 0.026), respectively. Overall, the average accuracy was 0.973 (SD: 0.026), suggesting that clustering results were highly consistent.

**Cluster 1: Elderly with mild cough.** Cluster 1 represented 17.3% (44/255) of the participants and consisted of elderly patients. Symptoms of cough were mildest, as the total COAT scores and scores for each question were the lowest, and all question scores for the K-LCQ were the highest. The proportion of patients with GERD was higher in cluster 1 than in the other clusters.

**Cluster 2: Middle-aged with less severe cough.** Cluster 2 represented 26.3% (67/255) of the participants and included a wide range of age groups, with middle-aged mean values. The total COAT scores and scores for each question were less severe than those of cluster 3 or 4, but more severe than those of cluster 1. Total K-LCQ scores and scores for each of its domains showed a similar pattern to that of the COAT scores. The scores for each question of the K-LCQ, except for questions 1, 2, 15, 16, and 17, were higher than the average K-LCQ scores. The proportion of asthma/CVA patients was the lowest in cluster 2.

**Cluster 3: Relatively male-predominant youth with severe cough.** Cluster 3 represented 31.8% (81/255) of the participants and comprised the youngest population. Although less than

**Table 1. Baseline characteristics of the entire study population and that compared between clusters of patients with chronic cough.**

|  | Total | Cluster 1 | Cluster 2 | Cluster 3 | Cluster 4 |
|---|---|---|---|---|---|
|  | (N = 255) | (N = 44) | (N = 67) | (N = 81) | (N = 63) |
| **Demographics** |  |  |  |  |  |
| Age | 47.7 ± 14.3 | 61.0 ± 11.1 | 41.3 ± 11.1* | 36.3 ± 7.4* | 60.1 ± 7.2 |
| Female sex | 161 (63.1%) | 28 (63.6%) | 37 (55.2%) | 43 (53.1%) | 53 (84.1%)* |
| Current smoker | 15 (5.9%) | 3 (6.8%) | 5 (7.5%) | 5 (6.2%) | 2 (3.2%) |
| Duration (week) | 18.3 ± 18.4 | 14.8 ± 11.7 | 19.2 ± 23.6 | 16.2 ± 12.4 | 24.3 ± 22.3 |
| **Cough severity** |  |  |  |  |  |
| **COAT total** | 11.3 ± 4.1 | 5.3 ± 2.1 | 8.9 ± 1.7* | 14.7 ± 2.0* | 13.8 ± 2.1* |
| COAT 1 | 2.6 ± 0.8 | 1.6 ± 0.8 | 2.2 ± 0.7 | 3.2 ± 0.4* | 2.9 ± 0.5* |
| COAT 2 | 2.2 ± 1.1 | 0.7 ± 0.7 | 1.9 ± 0.7* | 2.9 ± 0.7* | 2.6 ± 0.7* |
| COAT 3 | 1.7 ± 1.2 | 0.8 ± 1.0 | 1.0 ± 0.9 | 2.3 ± 1.1* | 2.5 ± 0.9* |
| COAT 4 | 2.0 ± 1.2 | 0.5 ± 0.7 | 1.3 ± 0.8* | 2.9 ± 0.6* | 2.6 ± 0.9* |
| COAT 5 | 2.8 ± 1.0 | 1.6 ± 1.0 | 2.4 ± 0.8 | 3.4 ± 0.7* | 3.1 ± 0.8* |
| **K-LCQ total** | 11.2 ± 3.1 | 14.5 ± 2.8 | 12.7 ± 2.3* | 9.2± 1.9* | 9.7 ± 2.7* |
| Physical | 4.1 ± 0.9 | 4.9 ± 1.0 | 4.5 ± 0.7* | 3.6 ± 0.6* | 3.6 ± 0.9* |
| Psychological | 3.5 ± 1.2 | 4.6 ± 1.0 | 4.0 ± 1.0* | 2.8 ± 0.8* | 3.0 ± 1.0* |
| Social | 3.6 ± 1.3 | 5.0 ± 1.1 | 4.2 ± 1.0* | 2.8 ± 0.9* | 3.1 ± 1.2* |
| **Cough NRA** | 6.0 ± 2.2 | 3.5 ± 1.8 | 5.2 ± 1.8* | 7.4 ± 1.6* | 6.9 ± 1.7* |
| **Diagnosis** |  |  |  |  |  |
| UACS | 116 (45.5%) | 17 (38.6%) | 30 (44.8%) | 39 (48.1%) | 30 (47.6%) |
| Asthma/CVA | 66 (25.9%) | 9 (20.5%) | 9 (13.4%)* | 24 (29.6%) | 24 (38.1%)* |
| EB | 41 (16.1%) | 9 (20.5%) | 13 (19.4%) | 13 (16.0%) | 6 (9.5%) |
| GERD | 34 (13.3%) | 11 (25.0%) | 14 (20.9%) | 6 (7.4%)* | 3 (4.8%)* |
| Idiopathic cough | 16 (6.3%) | 2 (4.5%) | 6 (9.0%) | 3 (3.7%) | 5 (7.9%) |
| Multiple cause | 20 (7.8%) | 4 (9.1%) | 4 (6.0%) | 6 (7.4%) | 6 (9.5%) |

* Indicates statistical significance compared to Cluster 1 as the reference group

COAT, cough assessment test; LCQ, Leicester cough questionnaire; NRS, numeric rating scale; UACS, upper airway cough syndrome; CVA, cough variant asthma; EB, eosinophilic bronchitis; GERD, gastroesophageal reflux disease

COAT 1: frequency of cough, COAT 2: limitation of daily activities, COAT 3: sleep disturbance, COAT 4: fatigue, COAT 5: hypersensitivity to irritants

50%, the proportion of men (46.9%) was the highest, and the total COAT scores and the scores for each COAT question were the highest. The total K-LCQ scores and scores for each K-LCQ domain were the lowest in this cluster. All questions of the K-LCQ, except question 9, had the lowest scores in cluster 3. The proportion of GERD patients was lower than cluster 1.

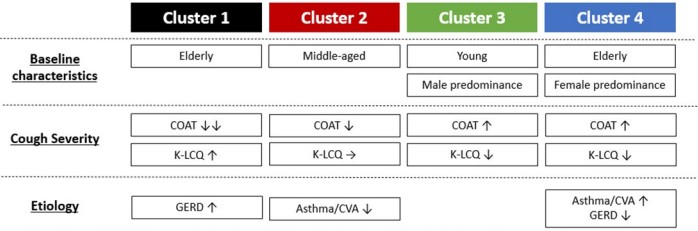

**Fig 3. Summary of baseline characteristics, cough severities, and aetiologies among clusters.** COAT, cough assessment test; K-LCQ, Korean version of Leicester cough questionnaire; CVA, cough variant asthma; GERD, gastroesophageal reflux disease.



**Fig 4. Summary of COAT scores among clusters.** COAT 1: Frequency of cough, COAT 2: Limitation on daily activities, COAT 3: Sleep disturbance, COAT 4: Fatigue, COAT 5: Hypersensitivity to irritants.

**Cluster 4: Female-predominant elderly with severe cough.** Cluster 4 represented 24.7% (63/255) of the participants and consist of elderly patients. The proportion of female patients was the highest in this cluster. The total COAT and K-LCQ scores were the most severe, similar to cluster 3. However, unlike cluster 3, the proportion of asthma/CVA was the highest, whereas that of GERD was the lowest in cluster 4.

## Discussion

We categorised the heterogeneous population with chronic cough into 4 clusters based on demographics, characteristics, and severity of the cough: elderly with mild cough, middle-aged with less severe cough, relatively male-predominant youth with severe cough, and female-predominant elderly with severe cough. We found that patients within each cluster exhibited varying characteristics and underlying diseases. These findings confirm the existence of heterogeneity in chronic cough and the need for improved phenotyping methods.

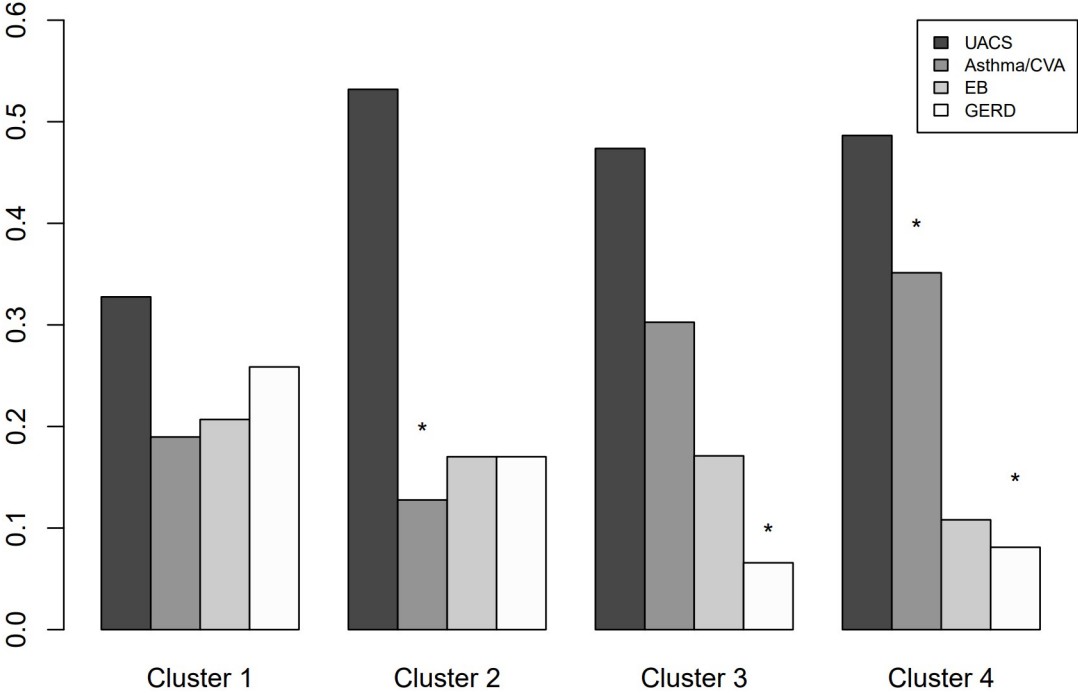

**Fig 5. Proportion of each cause according to clusters.** * Indicates statistical significance. Y axis means proportion of each etiology according to clusters (X axis). UACS, upper airway cough syndrome; CVA, cough variant asthma; EB, eosinophilic bronchitis; GERD, gastroesophageal reflux disease.

Although the mean age was not significantly different between each cause, age has been suggested as one of the important variables in dividing patients by aetiology of cough [20]. An increase in age was negatively correlated with the total COAT score in the overall sample; however, there was no such correlation within each cluster. Decrease in subjective perception of cough severity with increasing age suggests the possible influence of age on cough generation or perception. Female sex was positively associated with the score for COAT question 4 and the presence of asthma/CVA. Cough symptoms were the most severe in the asthma/CVA group but the mildest in the GERD group. However, the cough duration was not a distinct phenotype.

Cluster 1 consisted of the oldest population but showed the least severe cough. In this cluster, GERD contributed to a higher proportion of the underlying causes of cough than in the others. Cluster 2 was composed of a wide range of ages with medium cough severity and a middle-aged mean age value. This cluster had the lowest proportion of asthma/CVA cases. Although both cluster 3 and 4 consisted of patients with the most severe cough, different patterns in demographics and underlying diseases were observed. Cluster 3 had the youngest age group with relative male predominance; in contrast, cluster 4 was the oldest group, with female predominance. Additionally, the proportion of underlying diseases for cough was significantly different, since cluster 4 demonstrated the highest proportion of patients with asthma/CVA and the lowest proportion of those with GERD. Patients with chronic cough in cluster 4 need additional attention due to the high prevalence of asthma/CVA, and rapid diagnosis and treatment are required to prevent disease progression. Meanwhile, more empirical therapeutic trials for proton pump inhibitors could be applied to cluster 1 patients. These clusters could assist clinicians' decisions by categorising patients to predict underlying causes and indicate further required diagnostic procedures.

The prevalence of GERD was highest in cluster 1 (elderly with mild cough) and lowest in cluster 4 (female-predominant elderly with severe cough). Asthma was prevalent in cluster 4, but less prevalent in cluster 2 (middle-aged with less severe cough). The co-existence of GERD and asthma had been frequently reported [21, 22]. The prevalence of GERD is 21% in mild-to-moderate asthma, while it ranges from 46% to 63% in severe asthma [23]. It was suggested that negative pressure generated by airway obstruction may increase the pressure gradient between the chest and abdomen and provoke reflux. On the other hand, GERD can worsen asthma symptoms by aggravating airway hypersensitivity through aspiration-induced inflammation. Despite their accompaniment, we could see distinct pattern of clustering those diseases, which could help to differentiate the aetiology and understand the detailed pathophysiology of diseases. Moreover, patients with GERD reported less severe cough. One of our hypotheses is that typical GERD-related coughs tend to be periodic, such as coughing at night or after meals, so that the overall mean cough severity may be less severe than in diseases with persistent inflammation of the respiratory tract.

Cluster 3 (relatively male-predominant youth with severe cough) presented similar distribution of underlying causes with the total population. Since age and cough severity are different between each aetiology, characteristics of the common aetiology in each cluster could represent the characteristics of clusters. However, cluster 3 presented similar prevalence of diseases with the total population and could be used as an index group reflecting the characteristics of the general population with chronic cough rather than those of group with specific underlying diseases.

One strength of our study was that we described the characteristics of cough using a standard cough questionnaire measured quantitatively and utilised the results to categorise patients objectively. Moreover, clinicians could easily categorise patients into clusters in clinical practice, due to the simplicity of the COAT questionnaire. Furthermore, these data were

collected by pulmonologists from respiratory centres of academic teaching hospitals, especially those who specialise in chronic cough, following the guidelines for diagnosis and treatment of the condition. Therefore, the diagnosis of the underlying causes is reliable, with a low possibility of misdiagnosis.

This analysis demonstrates a novel finding regarding the subtyping of chronic cough. However, there are several limitations. Although these data were collected by respiratory specialists from academic teaching hospitals, detailed results of physical examination, spirometry, bronchoprovocation tests, eosinophil counts in induced sputum and blood, or exhaled fractions of nitric oxide were not collected; therefore, analysis was limited. Body mass index also plays a role in the pathophysiology of several diseases; however, we could not include that information in our analysis due to the lack of dataset. Though we followed the Korean cough guideline in diagnosis of aetiology, there could be bias caused by different manners between different respiratory centres. The number of patients included for clustering was relatively small, which might have diminished the significance between clusters. Furthermore, we had to fill out large number of questionnaires, so considerable number of patients were excluded because of incomplete data collection. Clustering was evaluated only in one country and would need to be validated to further generalise our results. Further large-scale studies are needed to confirm our findings, especially in different countries and ethnicities. Since we did not gather prognostic data, such as treatment success or recurrence, longitudinal studies using clustering methods are required to evaluate the implications of clusters. Lastly, distinct differences in UACS or EB in each cluster were not observed, and additional methods to specify these differences are required.

In conclusion, cluster analysis using demographics and the characteristics of cough categorised four distinct phenotypes of patients with chronic cough. Patients within each cluster varied considerably in terms of symptoms, severity, and underlying causes. Subgrouping patients with chronic cough into homogenous phenotypes could provide a stratified medical approach for individualising diagnostic and therapeutic strategies. Further studies are needed to validate these results.

## Supporting information

**S1 Table. Correlation matrix of each aetiology with COAT (A) and K-LCQ (B) scores.**
(DOCX)

**S2 Table. Detailed K-LCQ scores among clusters in patients with chronic cough.**
(DOCX)

**S1 Fig. Histogram of COAT questionnaire.**
(DOCX)

**S2 Fig. Prevalence of causes of chronic cough (A) and Venn Diagram (B).**
(DOCX)

**S3 Fig. Plots of silhouette width according to cluster number.**
(DOCX)

**S4 Fig. Distribution of each cluster and their correlations between age and COAT score.**
(DOCX)

**S1 Data.**
(ZIP)

## Author Contributions

**Conceptualization:** Jin Woo Kim, Hyeon-Kyoung Koo.

**Data curation:** So Hee Park, Hyung Koo Kang, Hye Kyeong Park, Ji-Yong Moon, Hyeon-Kyoung Koo.

**Formal analysis:** Jiyeon Kang, Woo Jung Seo, Jieun Kang, Minseok Seo, Hyeon-Kyoung Koo.

**Funding acquisition:** Hyeon-Kyoung Koo.

**Methodology:** Hyeon-Kyoung Koo.

**Project administration:** Hyeon-Kyoung Koo.

**Resources:** Ji-Yong Moon, Deog Kyeom Kim, Seung Hun Jang, Jin Woo Kim.

**Supervision:** Sung-Soon Lee, Deog Kyeom Kim, Seung Hun Jang, Jin Woo Kim.

**Visualization:** Hyeon-Kyoung Koo.

**Writing – original draft:** Jiyeon Kang, Woo Jung Seo, Hyeon-Kyoung Koo.

**Writing – review & editing:** Jiyeon Kang, Woo Jung Seo, Jieun Kang, So Hee Park, Hyung Koo Kang, Hye Kyeong Park, Sung-Soon Lee, Ji-Yong Moon, Deog Kyeom Kim, Seung Hun Jang, Jin Woo Kim, Minseok Seo, Hyeon-Kyoung Koo.

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
