## [Decision Letter · Decision Letter 0]

26 Dec 2022

PONE-D-22-25031Clinical Phenotypes of Chronic Cough Categorised by Cluster AnalysisPLOS ONE

Dear Dr. Koo,

Thank you for submitting your manuscript to PLOS ONE. After careful consideration, we feel that it has merit but does not fully meet PLOS ONE’s publication criteria as it currently stands. Therefore, we invite you to submit a revised version of the manuscript that addresses the points raised during the review process.

ACADEMIC EDITOR:

Further clarification is necessary regarding individual components of cough clusters and why it is important to divide cough into the clusters identified by the authors in the manuscript.

Do you think that your findings challenge current thinking about cough as a clinical symptom? The evidence presented must be strong enough to prove your case. Try to cite all the relevant work that would contradict your thinking and address it appropriately.

We look forward to receiving your revised manuscript.

Kind regards,

Bharat Bhushan Sharma, M.D.

Academic Editor

PLOS ONE

Journal Requirements:

Additional Editor Comments:

The manuscript has been reviewed by a set of reviewers.

Further clarification is necessary regarding individual components of cough clusters and why it is important to divide cough into the clusters identified by the authors in the manuscript.

Do you think that your findings challenge current thinking about cough as a clinical symptom? The evidence presented must be strong enough to prove your case. Try to cite all the relevant work that would contradict your thinking and address it appropriately.

Reviewers' comments:

Reviewer's Responses to Questions

**Comments to the Author**

1. Is the manuscript technically sound, and do the data support the conclusions?

Reviewer #1: Yes

Reviewer #2: Yes

Reviewer #3: Partly

2. Has the statistical analysis been performed appropriately and rigorously? 

Reviewer #1: Yes

Reviewer #2: Yes

Reviewer #3: Yes

3. Have the authors made all data underlying the findings in their manuscript fully available?

Reviewer #1: Yes

Reviewer #2: Yes

Reviewer #3: Yes

4. Is the manuscript presented in an intelligible fashion and written in standard English?

Reviewer #1: Yes

Reviewer #2: Yes

Reviewer #3: Yes

5. Review Comments to the Author

Reviewer #1: Following up you arguments was a little challenging it would be better that you clear delineate the cough assessment and clustering. From table 1 female predominate all groups but this is not reflected in the text. There is some attempt to link clustering with etiology but other than Asthma/CVA and GERD its not claear if other clustering predicted cause.

This is a good start as the world is moving to ML and AI

Reviewer #2: This manuscript by Kang et al. focused on the clinical phenotypes of chronic cough. Authors evaluated the characteristics and severity of chronic cough using the COugh Assessment Test (COAT) and the Korean version of the Leicester Cough Questionnaire (K-LCQ) and categorized patients with chronic cough into 4 clusters using the cluster analysis. As authors mentioned, etiology of cough and underlying diseases in patients with chronic cough are various. Therefore, the concept of this study to clarify the characteristics of these patients is valuable and results seem agreeable. Although this manuscript seems written well, authors may want to consider several issues as follows.

Major comments

1) It is difficult to understand why the etiology of chronic cough was diagnosed only 4 diseases such as upper airway cough syndrome (UACS), asthma/cough variant asthma (CVA), eosinophilic bronchitis (EB), and gastroesophageal reflux disease (GERD). How did authors deal with other respiratory diseases such as interstitial pneumonia, COPD, and lung cancer? Were these diseases included in UACS? Otherwise, authors should clarify what diseases are included in UACS.

2) In relation to above, although only GERD was listed as an extra pulmonary cause of chronic cough, how did authors categorize other diseases such as chronic sinusitis and congestive heart failure.

3) It is unknown whether this clustering was associated with any outcome in patients with chronic cough from this study.

Minor comments

1) Number of patients seems relatively small to identify clusters.

2) In Table 1, “current smoker” is duplicated and is not a diagnosis. In addition, what does “idiopathic” mean? Is this same as “unknown cause”?

3) In Table 1, If so, its number may be 19 as described in the main text. Furthermore, according to the Supplementary Figure S2B, number of multiple cause should be 20. Please make sure all numbers carefully again.

4) In Table 1, for what dis asterisk indicate statistical significance? Comparison should be performed among 4 clusters.

Reviewer #3: In the manuscript, the authors tried to categorise the phenotypes of chronic cough by cluster analy based on the characteristics and severity of cough assessed by the COugh Assessment Test (COAT) and the Korean version of the Leicester Cough Questionnaire. They found chronic cough could be divided into four cluster pehenotypes: 1) elderly with mild cough, 2) middle-aged with less severe cough, 3) relatively male-predominant youth with severe cough, and 4) female-predominant elderly with severe cough. All clusters had distinct demographic and symptomatic characteristics and underlying causes. They concluded four distinct phenotypes of chronic cough reflected the significant differences in the aetiologies and provided a stratified medical approach for individualising diagnostic and therapeutic strategies. It is an interesting research. However, there are several isssus to be addressed.

Major comments

1.Four cough phenotypes were clustered based on the simple characteristics and simple tools of cough evaluation tool, and may reflect some feature of common etiologies underlying chronic cough. However, the common causes of chronic cough can be easily identified and management following the current approach for chronic cough. Do we really need these cluster phenotypes since their identification seems to have no potential ability to improve the diagonosis and treatment of these common etiologies?

2.In the cohort of the patients with chronic cough, there were 19 (6.3%) patients whose cause was not identified and 20 participants (7.8%) who had multiple causes for their cough. What about their distribution in four cluster phenotypes? I think it may help to seek specific therapy if the phenotypes of these chronic refractory cough can be identified by the cluster analysis because their management is difficult and challenging.

.

The minor comments

1.Cluster 1 had a mild cough and older age with GERD as a main underlying etiology. Please explain why GERD-associated cough coughed mildly.

2.It is a surprise that age was negatively correlated with the COAT scores for questions 1-5 and the total COAT.

6. PLOS authors have the option to publish the peer review history of their article (what does this mean?). If published, this will include your full peer review and any attached files.

Reviewer #1: **Yes: **Dr Evans Amukoye

Reviewer #2: No

Reviewer #3: No

---

## [Author Response · Author response to Decision Letter 0]

24 Jan 2023

Additional Editor Comments:

The manuscript has been reviewed by a set of reviewers.

Further clarification is necessary regarding individual components of cough clusters and why it is important to divide cough into the clusters identified by the authors in the manuscript.

Do you think that your findings challenge current thinking about cough as a clinical symptom? The evidence presented must be strong enough to prove your case. Try to cite all the relevant work that would contradict your thinking and address it appropriately.

Answer: Thank you for your comment. Cough is the most prevalent symptom of numerous diseases, including pulmonary and extra-pulmonary diseases. In addition, cough-related symptoms are highly variable, making it quite difficult to determine the underlying cause. Furthermore, guidelines for chronic cough evaluation require a variety of specialized equipment that is impractical for many primary care centers with limited resources, thereby increasing the difficulty for physicians. Due to these challenges, some guidelines also recommend empirical treatment for potential causes first, followed by re-evaluation of treatment response as an alternative strategy. Therefore, identifying the subtypes of these heterogeneous phenotypes of chronic cough is crucial for better understanding of the pathophysiology and treatment of disease. So far, the majority of research on chronic cough had primarily focused on epidemiology and diagnostic flow. Few studies had described phenotypes and compared differences according to etiology. The aim of our study was to identify specific subtypes using cluster analysis in order to assist physicians in diagnosing the cause of chronic cough based on the phenotypes of heterogeneous patients.  

Reviewers' comments:

5. Review Comments to the Author

Reviewer #1: Following up you arguments was a little challenging it would be better that you clear delineate the cough assessment and clustering. From table 1 female predominate all groups but this is not reflected in the text. There is some attempt to link clustering with etiology but other than Asthma/CVA and GERD its not clear if other clustering predicted cause.

This is a good start as the world is moving to ML and AI

Answer: Thank you very much for your comments. As you pointed, proportion of females in all clusters was greater than fifty percent, which is usual pattern for chronic cough population. We added the following sentences to the Result section to clarify the predominance of females. 

(Page 9, line 141) Among them, 94 (36.9%) men and 161 (63.1%) women were included, with a male-to-female ratio of 1:1.71. 

(Page 12, line 205) Cluster 3 represented 31.8% (81/255) of the participants and comprised the youngest population. Although less than 50%, the proportion of men (46.9%) was the highest, and the total COAT scores and the scores for each COAT question were the highest.

Furthermore, as you pointed, we were unable to identify the specific predominance other than asthma/CVA and GERD. To clarify the facts, we added the following limitation to Discussion section:

(Page 16, Line 296) Lastly, distinct differences of UACS or EB in each cluster were not observed, and additional methods to specify these differences are required.

 

Reviewer #2: This manuscript by Kang et al. focused on the clinical phenotypes of chronic cough. Authors evaluated the characteristics and severity of chronic cough using the COugh Assessment Test (COAT) and the Korean version of the Leicester Cough Questionnaire (K-LCQ) and categorized patients with chronic cough into 4 clusters using the cluster analysis. As authors mentioned, etiology of cough and underlying diseases in patients with chronic cough are various. Therefore, the concept of this study to clarify the characteristics of these patients is valuable and results seem agreeable. Although this manuscript seems written well, authors may want to consider several issues as follows.

Major comments

1) It is difficult to understand why the etiology of chronic cough was diagnosed only 4 diseases such as upper airway cough syndrome (UACS), asthma/cough variant asthma (CVA), eosinophilic bronchitis (EB), and gastroesophageal reflux disease (GERD). How did authors deal with other respiratory diseases such as interstitial pneumonia, COPD, and lung cancer? Were these diseases included in UACS? Otherwise, authors should clarify what diseases are included in UACS.

Answer: Thank you for your insightful comment. Previous reports have highlighted UACS, asthma including CVA, EB, and GERD as the primary causes of chronic cough in non-smokers with normal chest radiographs. We added the following sentence to the Introduction section:

(Page 4, Line 62) The most common causes of chronic cough are known as upper airway cough syndrome (UACS), cough variant asthma (CVA), eosinophilic bronchitis (EB), and gastroesophageal reflux disease (GERD) [3-6].

We also included patients without structural lung disease in chest X-ray or CT, following the previous definition. We appreciate for allowing us to clarify the inclusion criteria. Inclusion criteria in the Method section were specified as follows:

(Page 6, Line 88) The cause of chronic cough was assessed following the Korean cough guideline [18] by pulmonary specialists in each hospital, excluding those for patients with suspected abnormalities on plain chest radiography/computed tomography or with chronic respiratory disease, such as overt asthma, COPD, bronchiectasis, tuberculosis-destroyed lung, and lung cancer. 

2) In relation to above, although only GERD was listed as an extra pulmonary cause of chronic cough, how did authors categorize other diseases such as chronic sinusitis and congestive heart failure.

Answer: Thank you for your points. Chronic sinusitis was categorized as upper airway cough syndrome, while structural diseases with abnormal chest X-ray including heart failure were excluded. Additionally, only patients with chronic cough as the primary reason for hospital visit were enrolled; those with other diseases who primarily complained of other symptoms, such as dyspnea, but cough as a secondary symptom were not enrolled. We clarified the inclusion criteria in the Method section as above mentioned. 

(Page 6, Line 88) The cause of chronic cough was assessed following the Korean cough guideline [18] by pulmonary specialists in each hospital, excluding those for patients with suspected abnormalities on plain chest radiography/computed tomography or with chronic respiratory disease, such as overt asthma, COPD, bronchiectasis, tuberculosis-destroyed lung, and lung cancer. 

3) It is unknown whether this clustering was associated with any outcome in patients with chronic cough from this study.

Answer: Thank you for your comments. As you pointed, we did not collect prognostic data, and the lack of longitudinal data, such as treatment success, recurrence, or even death, is one of our study’s significant limitations. This limitation was outlined in the Discussion section. We modified this sentence further to specify the outcomes as follows:

(Page 16, Line 294) Since we did not gather prognostic data, such as treatment success or recurrence, longitudinal studies using clustering methods are required to evaluate the implications of clusters.

Minor comments

1) Number of patients seems relatively small to identify clusters.

Answer: Thank you for your points. That is correct. We added that limitation to the Discussion section as follows:

(Page 16, Line 288) The number of patients included for clustering was relatively small, which might have diminished the significance between clusters.

2) In Table 1, “current smoker” is duplicated and is not a diagnosis. In addition, what does “idiopathic” mean? Is this same as “unknown cause”?

Answer: Thank you very much for allowing us the opportunity to correct our error. The duplicated smoking variable was eliminated from Table 1. According to the ACCP guideline (Unexplained (idiopathic) cough: ACCP evidence-based clinical practice guidelines. Chest 2006 Jan;129(1 Suppl):220S-221S), idiopathic or unexplained cough was defined as a cough with no etiology identified after evaluation, also known as an unknown cause. To specify the cause, we changed the term ‘idiopathic’ to ‘idiopathic cough’ in Table 1.

3) In Table 1, If so, its number may be 19 as described in the main text. Furthermore, according to the Supplementary Figure S2B, number of multiple cause should be 20. Please make sure all numbers carefully again.

Answer: Thank you very much for thorough review and suggestions. We sincerely appreciate the opportunity to correct our error. As you pointed, the exact number of patients with multiple causes is 20 and those with idiopathic cough is 16. Changes are made to the numbers in the Result section and Table 1. 

(Page 9, Line 148) In 16 (6.3%) patients, the cause could not be identified and a total of 20 participants (7.8%) had multiple causes for their cough (S2B Fig).

4) In Table 1, for what dis asterisk indicate statistical significance? Comparison should be performed among 4 clusters.

Answer: We greatly appreciate your comment. All comparisons were re-evaluated using cluster 1 as the reference. 

 

Reviewer #3: In the manuscript, the authors tried to categorise the phenotypes of chronic cough by cluster analy based on the characteristics and severity of cough assessed by the COugh Assessment Test (COAT) and the Korean version of the Leicester Cough Questionnaire. They found chronic cough could be divided into four cluster phenotypes: 1) elderly with mild cough, 2) middle-aged with less severe cough, 3) relatively male-predominant youth with severe cough, and 4) female-predominant elderly with severe cough. All clusters had distinct demographic and symptomatic characteristics and underlying causes. They concluded four distinct phenotypes of chronic cough reflected the significant differences in the aetiologies and provided a stratified medical approach for individualising diagnostic and therapeutic strategies. It is an interesting research. However, there are several issues to be addressed.

Major comments

1.Four cough phenotypes were clustered based on the simple characteristics and simple tools of cough evaluation tool, and may reflect some feature of common etiologies underlying chronic cough. However, the common causes of chronic cough can be easily identified and management following the current approach for chronic cough. Do we really need these cluster phenotypes since their identification seems to have no potential ability to improve the diagnosis and treatment of these common etiologies?

Answer: Thank you for your comments. In guideline, algorithm for chronic cough evaluation includes chest radiography, PNS view, spirometry with bronchodilator reversibility test, bronchoprovocation test, induced sputum analysis, FENO, gastro-endoscopy, and for the specific diagnosis, chest CT, PNS CT, bronchoscopy, and echocardiography are additionally required. However, these tests require a variety of specialized laboratory equipment and space, which is not feasible for many primary care centers due to limited resources. Because of these challenges, some guidelines also recommend empirical treatment for potential causes first and subsequent re-evaluating treatment response as an alternative strategy. Therefore, clustering the phenotype would be advantageous for many primary care centers that treat numerous patients with chronic cough. We added the following sentence to the Introduction section: 

(Page 4, Line 64) However, the associated symptoms of chronic cough vary widely, and determining the aetiology of chronic cough is a difficult process as it can develop for various reasons [7-11]. Guidelines for the evaluation of chronic cough include a variety of specialized equipment, which is not feasible for many primary care centres due to limited resources [3-6].

2.In the cohort of the patients with chronic cough, there were 19 (6.3%) patients whose cause was not identified and 20 participants (7.8%) who had multiple causes for their cough. What about their distribution in four cluster phenotypes? I think it may help to seek specific therapy if the phenotypes of these chronic refractory cough can be identified by the cluster analysis because their management is difficult and challenging.

Answer: Thank you for your insightful comment. Unfortunately, there were no significant differences in the distribution of these patients between clusters. The numbers and percentages are detailed in Table 1. 

The minor comments

1.Cluster 1 had a mild cough and older age with GERD as a main underlying etiology. Please explain why GERD-associated cough coughed mildly.

Answer: Thank you very much for your comment. We were unable to find the reference because trials comparing cough severity in each etiology had not been conducted. One of our potential hypotheses is that typical GERD-related coughs tend to be periodic, such as coughing at night time or after meal, so that the overall mean cough severity may be less severe than in other diseases with persistent respiratory tract inflammation. We added following explanation to the Discussion section:

(Page 14, Line 260) Moreover, patients with GERD reported less severe cough. One of our hypotheses is that typical GERD-related coughs tend to be periodic, such as coughing at night or after meal, so that the overall mean cough severity may be less severe than in diseases with persistent inflammation of respiratory tract. 

2. It is a surprise that age was negatively correlated with the COAT scores for questions 1-5 and the total COAT.

Answer: Thank you very much for your comment. We added following sentences to explain the potential mechanism. 

(Page 13, Line 228) An increase in age was negatively correlated with the total COAT score in the overall sample; however, there was no such correlation within each cluster. Decrease of subjective perception of cough severity with increasing age suggests the possible influence of age on generation or perception of cough.

---

## [Decision Letter · Decision Letter 1]

2 Mar 2023

PONE-D-22-25031R1Clinical phenotypes of chronic cough categorised by cluster analysisPLOS ONE

Dear Dr. Koo,

Thank you for submitting your manuscript to PLOS ONE. After careful consideration, we feel that it has merit but does not fully meet PLOS ONE’s publication criteria as it currently stands. Therefore, we invite you to submit a revised version of the manuscript that addresses the points raised during the review process.

ACADEMIC EDITOR:

Please proofread your paper thoroughly and correct spelling, punctuation, grammar, and formatting errors. Because all references will be linked electronically as much as possible to the papers they cite, proper formatting of references is crucial. For more information, please refer to submission guidelines of the journal.

We look forward to receiving your revised manuscript.

Kind regards,

Bharat Bhushan Sharma, M.D.

Academic Editor

PLOS ONE

Journal Requirements:

Additional Editor Comments:

Please proofread your paper thoroughly and correct spelling, punctuation, grammar, and formatting errors. Because all references will be linked electronically as much as possible to the papers they cite, proper formatting of references is crucial. For more information, please refer to submission guidelines of the journal.

Reviewers' comments:

Reviewer's Responses to Questions

**Comments to the Author**

1. If the authors have adequately addressed your comments raised in a previous round of review and you feel that this manuscript is now acceptable for publication, you may indicate that here to bypass the “Comments to the Author” section, enter your conflict of interest statement in the “Confidential to Editor” section, and submit your "Accept" recommendation.

Reviewer #1: All comments have been addressed

Reviewer #2: All comments have been addressed

Reviewer #3: All comments have been addressed

2. Is the manuscript technically sound, and do the data support the conclusions?

Reviewer #1: Yes

Reviewer #2: Yes

Reviewer #3: Yes

3. Has the statistical analysis been performed appropriately and rigorously? 

Reviewer #1: Yes

Reviewer #2: Yes

Reviewer #3: Yes

4. Have the authors made all data underlying the findings in their manuscript fully available?

Reviewer #1: Yes

Reviewer #2: Yes

Reviewer #3: Yes

5. Is the manuscript presented in an intelligible fashion and written in standard English?

Reviewer #1: Yes

Reviewer #2: Yes

Reviewer #3: Yes

6. Review Comments to the Author

Reviewer #1: This is a good start In using data (AI) to predict causes of chronic cough. I note no clustering in certain groups that leaves a lacuna, but as we get more data we may find out if we may/may not cluster all possible groups.

I accept your responses

Reviewer #2: This revised manuscript by Kang et al. focused on the clinical phenotypes of chronic cough. Authors evaluated the characteristics and severity of chronic cough using the COugh Assessment Test (COAT) and the Korean version of the Leicester Cough Questionnaire (K-LCQ) and categorized patients with chronic cough into 4 clusters using the cluster analysis. Authors revised the manuscript appropriately according to the reviewers’ comments. It appeared better. I do not have further comment to be resolved.

Reviewer #3: After revision, the quality of manuscript has greatly been improved. I have no further comments for the revised manuscript that need to be addressed.

7. PLOS authors have the option to publish the peer review history of their article (what does this mean?). If published, this will include your full peer review and any attached files.

Reviewer #1: No

Reviewer #2: No

Reviewer #3: No

---

## [Author Response · Author response to Decision Letter 1]

3 Mar 2023

Thank you very much for allowing us the opportunity to correct our errors. We corrected spelling, punctuation, grammar, and reference formatting errors all through the manuscript. Both tracked and clean versions of the manuscript are provided. Again, we appreciate all the invaluable comments that improved the overall quality of our manuscript.

---

## [Editor Report · Decision Letter 2]

7 Mar 2023

Clinical phenotypes of chronic cough categorised by cluster analysis

PONE-D-22-25031R2

Dear Dr. Koo,

We’re pleased to inform you that your manuscript has been judged scientifically suitable for publication and will be formally accepted for publication once it meets all outstanding technical requirements.

Kind regards,

Bharat Bhushan Sharma, M.D.

Academic Editor

PLOS ONE

---

## [Editor Report · Acceptance letter]

9 Mar 2023

PONE-D-22-25031R2 

Clinical phenotypes of chronic cough categorised by cluster analysis 

Dear Dr. Koo:

I'm pleased to inform you that your manuscript has been deemed suitable for publication in PLOS ONE. Congratulations! Your manuscript is now with our production department. 

Kind regards, 

on behalf of

Professor Bharat Bhushan Sharma 

Academic Editor

PLOS ONE